# Activation/Inhibition of Cholinesterases by Excess Substrate: Interpretation of the Phenomenological *b* Factor in Steady-State Rate Equation

**DOI:** 10.3390/ijms241310472

**Published:** 2023-06-21

**Authors:** Aliya R. Mukhametgalieva, Andrey V. Nemtarev, Viktor V. Sykaev, Tatiana N. Pashirova, Patrick Masson

**Affiliations:** 1Biochemical Neuropharmacology Laboratory, Kazan Federal University, 18 Ul. Kremlevskaya, 420008 Kazan, Russia; aliya_rafikovna@mail.ru; 2Arbuzov Institute of Organic and Physical Chemistry, Kazan Scientific Center, Russian Academy of Sciences, 8 Ul. Arbuzov, 420088 Kazan, Russia; a.nemtarev@mail.ru (A.V.N.); vsyakaev@iopc.ru (V.V.S.); tatyana_pashirova@mail.ru (T.N.P.)

**Keywords:** acetylcholinesterase, butyrylcholinesterase, excess substrate activation/inhibition, *b* factor, competing substrate kinetics

## Abstract

Cholinesterases (ChEs) display a non-michaelian behavior with positively charged substrates. In the steady-state rate equation, the b factor describes this behavior: if *b >* 1 there is substrate activation, if *b <* 1 there is substrate inhibition. The mechanistic significance of the *b* factor was investigated to determine whether this behavior depends on acylation, deacylation or on both steps. Kinetics of human acetyl- (AChE) and butyryl-cholinesterase (BChE) were performed under steady-state conditions and using a time-course of complete substrate hydrolysis. For the hydrolysis of short acyl(thio)esters, where acylation and deacylation are partly rate-limiting, steady-state kinetic analysis could not decide which step determines *b*. However, the study of the hydrolysis of an arylacylamide, 3-(acetamido)-N,N,N-trimethylanilinium (ATMA), where acetylation is rate-limiting, showed that *b* depends on the acylation step. The magnitude of *b* and opposite *b* values between AChE and BChE for the hydrolysis of acetyl(thio)- versus benzoyl-(thio) esters, then indicated that the productive adjustment of substrates in the active center at high concentration depends on motions of both the Ω and the acyl-binding loops. Benzoylcholine was shown to be a poor substrate of AChE, and steady-state kinetics showed a sudden inhibition at high concentration, likely due to the non-dissociation of hydrolysis products. The poor catalytic hydrolysis of this bulky ester by AChE illustrates the importance of the fine adjustment of substrate acyl moiety in the acyl-binding pocket. Molecular modeling and QM/MM simulations should definitively provide evidence for this statement.

## 1. Introduction

Cholinesterases (ChEs) are widely distributed in living organisms [1]. In animals, the main function of acetylcholinesterase (AChE, EC. 3.1.1.7) is to terminate the action of the neutrotransmitter acetylcholine (ACh) in the central nervous system (CNS), ganglions and at neuromuscular junctions (NMJ). In addition, AChE has non-cholinergic functions in cellular development [2]. The physiological functions of butyrylcholinesterase (BChE, EC. 3.1.1.8) are still debated. BChE may serve as a surrogate for AChE or as its backup under extreme physiological conditions, but likely also plays a constitutive role in the brain under normal conditions [3]. Other possible functions have been proposed. In particular, BChE deacylates ghrelin, the “hunger” hormone, an activity that may be relevant in fatty acid metabolism [4]. It was known for a long time that BChE hydrolyzes long-chain acylcholines, and recently it was shown that this property could modulate inflammatory processes such as the “cytokine storm” observed in severe forms of COVID-19 [5]. Furthermore, BChE displays a wide specificity towards esters. This includes ester-containing drugs such as aspirin [6], the myorelaxant succinylcholine [7], and poisonous carboxyl-esters like heroin and cocaine [8]. Both ChEs also display a promiscuous activity in hydrolyzing arylacylamides [9,10].

Both enzymes are irreversibly inhibited by poisonous organophosphates and carbamates, causing major cholinergic syndrome [11]. Thus, endogenous BChE is of toxicological importance in reacting with poisonous carbamyl-esters and phosphoryl-esters [12,13]. Endogenous BChE exerts protection of AChE in the cholinergic system in metabolizing or scavenging esters of natural or artificial origin. In particular, it was established that the administration of highly purified human BChE is an effective stoichiometric bioscavenger against the acute toxicity of organophosphate pesticides and nerve agents [12,14,15]. The inhibition (reversible and/or irreversible) of both ChEs has also been used in the treatment of Alzheimer’s disease [16]. The current and potential therapeutic uses of BChE in detoxification, in the treatment of obesity and in Alzheimer’s disease have recently been reviewed [17].

The 3D structure of human ChEs recombinant monomer was determined by X ray diffraction analysis (PDB 1PO1 for BChE and 4EY4 for AChE) [18,19] and that of naturally-occurring human plasma BChE tetramer by cryo-electron microscopy [20,21]. The 3D structure of membrane-anchored oligomeric forms of both ChEs has not yet been solved [22]. Monomers of both enzymes are single domain α/β fold proteins. In tetrameric structures, monomers are equivalent and show no cooperativity. In each monomer, the catalytic active site (CAS) is located at the bottom of a deep narrow gorge of 20 Å, just in the middle of the enzyme core (Figure 1). The main component of the CAS is the catalytic triad Ser-His-Asp (S198-H438-E325 in human BChE) surrounded by an oxyanion hole, plus a glutamate (D) adjacent to the catalytic serine (S). The protonation of this glutamate plays a key role in stabilizing the catalytic triad [23]. On both sides of the catalytic serine there are two binding sites: a π-cation binding site and an acyl-binding site. At the entrance of the active site gorge a peripheral anionic site (PAS) is located, which is the recognition binding site of substrates and ligands [18,19,24]. These molecules bind transiently to the PAS, and then slide down the gorge to the CAS. PAS and CAS are interconnected through the acyl-binding loop and the Ω loop (Figure 1). In AChE (D74 and Y334), as well as in BChE (D70 and Y332), part of these two loops are H-bonded at the rim of the active site gorge. The Ω loop interconnects the main residue of the PAS (D70 in human BChE, D74 in AChE) and the tryptophan (W) of the CAS (W82 in human BChE, W86 in AChE). This tryptophan establishes π-cation interactions with positively charged substrates and ligands. Therefore, any conformational change in the Ω loop induces a change in the binding of substrates and inhibitors in the CAS. The acyl-binding pocket loop interconnects the PAS with the acyl-binding pocket (ABP). The importance of this loop for the productive binding of substrate acyl moiety has long been recognized [25,26,27]. Recent crystallographic and small angle X-ray scattering (SAXS) data on AChE conjugates confirmed the conformational plasticity of this loop for the adaptative binding of various ligands, substrates and covalent inhibitors [28,29]. As a result, ChEs display a complex catalytic behavior particularly with positively charged substrates. The events determining this behavior are not completely understood. With neutral esters, the hydrolysis of substrates obeys the simple two-step Michaelis–Menten model. After the formation of the reversible michaelian complex ES, the active site serine is acylated (*k*_2_), and subsequently deacylated (*k*_3_), by the nucleophilic attack of water, acting as a co-substrate (Figure 1).

As a result, ChEs display a complex catalytic behavior particularly with positively charged substrates. The events determining this behavior are not completely understood. With neutral esters, the hydrolysis of substrates obeys the simple two-step Michaelis–Menten model. After the formation of the reversible michaelian complex ES, the active site serine is acylated (*k*_2_), and subsequently deacylated (*k*_3_), by the nucleophilic attack of water, acting as a co-substrate (Figure 1).

In Figure 1, ES is the productive enzyme–substrate complex, EA is the acylated enzyme, P_1_ is the alcohol/phenol product and P_2_ is the acid product. For ester substrates, acylation and deacylation are partially rate-limiting (*k*_2_ ≥ *k*_3_) [9,30,31]. On the other hand, in the case of poor substrates, like arylacylamides, *k*_2_
*<< k*_3_, then *k_cat_* = *k*_2_ and *K_m_* = *K_s_* [9,10]. The kinetic Figure 1 is described by the classical Michaelis–Menten rate equation (Equation (1)):(1)$$v=\frac{k_{cat}\left[E\right]\left[S\right]}{K_{m}+\left[S\right]}$$
where
(2)$$k_{cat}=\frac{V_{max}}{\left[E\right]}=\frac{k_{2}k_{3}}{k_{2}+k_{3}}$$
and
(3)$$K_{m}=\frac{\left(k_{-1}+k_{2}\right)k_{3}}{k_{1}\left(k_{2}+k_{3}\right)}=\frac{K_{s}k_{3}}{\left(k_{2}+k_{3}\right)}=\frac{K_{s}}{\left(1+\frac{k_{2}}{k_{3}}\right)}\;$$

In Equation (3), *K_s_ = (k*_−1_
*+ k*_2_*)/k*_1_ is the dissociation of the enzyme–substrate complex ES. However, with positively charged substrates, such as the natural substrate acetylcholine and, in fact, with the majority of known ChE substrates, ChEs deviate from the michaelian behavior at high substrate concentrations, being either activated or inhibited by excess substrate. With these substrates, the minimum catalytic mechanism is conveniently described by the Webb model (Figure 2) [25]:

In Figure 2, at high substrate concentrations, a second substrate molecule (S_p_) binds to the (PAS, p), giving a ternary complex, S_p_ES, characterized by a dissociation constant *K_ss_*. The binding of this second substrate molecule to the PAS triggers an allosteric effect via motions of both the Ω loop and the acyl-binding loop (Figure 1), causing an alteration in the catalytic constant *k_cat_* by a *b* factor. If *b* < 1, there is inhibition by excess substrate; on the contrary, if *b* > 1, there is activation by excess substrate. If *b* = 1, the catalytic behavior is michaelian. The non-michaelian behavior of ChEs is conveniently described by Equation (4), popularized by Radic [26]. This equation is now used by most researchers working on catalytic and inhibition mechanisms of cholinesterases.
(4)$$v=\left(\frac{k_{cat}\left[E\right]}{1+K_{m}/\left[S\right]}\right)\left(\frac{1+b\left[S\right]/K_{ss}}{1+\left[S\right]/K_{ss}}\right)$$

As seen here, at low substrate concentration where [S] << *K_ss_*, => *b* → 1, Figure 2 reduces to the simple Michalis–Menten model (Figure 1) described by Equation (1). However, several issues remain unsolved. In particular, it is unclear whether the binding of a second substrate molecule (or a ligand) on the PAS affects acylation or deacylation. Thus, the *b* factor in Figure 2 is essentially an overall phenomenological parameter.

The goal of the present work was to determine which catalytic step(s) determine the *b* factor. The *b* factor may indeed result from two additive contributions, *a* and *d*, acting on acylation (*a*) and deacylation (*d*), respectively. For this purpose, steady-state kinetics was performed, using (thio)esters and a positively charged arylacetylamide (ATMA). It was shown that the « *a* » contribution is the sole determinant of the catalytic behavior of ChEs at high substrate concentration, causing either an activation or inhibition by excess substrate with charged substrates. An analysis of the steady-state AChE-catalyzed hydrolysis of a bulky ester (BzCh), and of progress curves for competition between BzCh and BzTC, showed that the heteroatom (O vs. S) affects the productive binding and *b*. This indicates that the fine adjustment of the benzoyl moiety in the ABP results from the functional flexibility of this loop, making AChE capable of accommodating bulky substrates.

## 2. Results

### 2.1. Steady-State Kinetics of AChE and BChE with ATMA

The BChE- and AChE-catalyzed hydrolysis of the arylacetylamide substrate ATMA is non-michaelian, showing activation by excess substrate up to 6 mM, and then inhibition for positively charged (thio)esters at very high concentrations (Figure 2). From the phase of activation by excess substrate, *b* was calculated using Equation (4). The *b* values are 1.8 ± 0.4 and 3.1 ± 0.6 for BChE and AChE, respectively. Because for this substrate, acylation is the rate-limiting step (*k*_2_ << *k*_3_) for both enzymes; Equation (9) is equal to 0 with *a* − *b* = 0, *d* = 1 and *a* = *b*. At high substrate concentrations up to 6 mM (Equation (8)), *b∙k*_cat_ = *a∙k*_2_. Acylation is the sole contributor to the *b* factor, i.e., *b* = *a.* For both enzymes, the inhibitory phase, beyond 6 mM ATMA may correspond to product inhibition, forming an abortive complex SEP_2_. Such an inhibitory phase at very high substrate concentrations has been observed with all positively charged substrates of ChEs but has never, thus far, been thoroughly investigated.

### 2.2. Steady-State Hydrolysis of BzCh and BzTC by Human AChE

The human AChE-catalyzed hydrolysis of BzTC was found to be very slow and strongly activated by excess substrate (Figure 3). Data were fitted to Equation (4), giving *K_m_* = 0.32 ± 0.03 mM; *K_ss_* = 1.34 ± 0.4 mM; *k_cat_* = 18 ± 7 min^−1^ and *b* = 7.8 ± 0.5. These results are in agreement with rates reported by Hosea et al. [27] for mouse AChE (Table 1, footnote *b*).

BzCh was also found to be a poor substrate of human AChE with *K_m_* = 0.3 ± 0.07 mM and *k_cat_* = 72 ± 4 min^−1^. This is in agreement with previously reported studies (Table 1). Although the *b* factor for the AChE-catalyzed hydrolysis of BzTC hydrolysis is high, it was impossible to determine the *b* factor for the hydrolysis of BzCh. Indeed, the steady-state AChE-catalyzed hydrolysis of BzCh, up to 800 μM, revealed unusual behavior beyond 500 μM (Figure 4A): a rapid drop of activity within a narrow interval of BzCh concentration. This behavior does not fit with the model described in Figure 2 and Equation (4).

After conversion, the initial rates, ranging from 0.0016 to 0.0082 ΔA_240_/min, were expressed in terms of μmol./min of released product: −*d*[*BzCh*]/*dt* = *d*[*P*_1_]/*dt = d*[*P*_2_]/*dt.*

A statistical analysis of the residuals was used to discriminate between the two models that describe the catalytic mechanisms of ChEs (Figure 1 and Figure 2). For this purpose, we compared the sum of square Q^2^ (for details, see Appendix A). Plots of residuals, as a function of predicted velocities, were calculated by fitting the experimental data of Figure 4 to the Michaelis–Menten (Figure 1), and Webb (Figure 2) models are reported (Appendix A). The narrowly scattered distribution of residuals around the horizontal axis, up to 600 μM BzCh, indicates that the kinetics obey the Michaelis–Menten model up to this concentration. However, the use of the Michaelis–Menten model and the Webb–Radic model, for BzCh concentrations up to 800 μM, shows an abnormal distribution of residuals (Appendix A). Errors in estimates of rates are not the result of experimental measurements but reveal a change in the catalytic behavior. In particular, the sudden collapse of residuals above 600 μM indicates that the kinetics follows neither of the two Schemes beyond this concentration.

To interpret this phenomenon, several hypotheses were considered. Firstly, because of the structural analogy of BzCh with cationic denaturing agents, such as benzalkonium, we speculated whether concentrations of BzCh > 600 μM could induce the unfolding of AChE. ChEs are very sensitive to chemical denaturants and undergo *irreversible* denaturation in the presence of such agents [35]. In fact, after the microdialysis or dilution of AChE samples subjected to high concentrations of BzCh, the catalytic activity was fully recovered. Thus, BzCh does not denature AChE. Then, two alternative hypotheses were proposed: (a) at high BzCh concentration, substrate molecules arrange as dimer and multimers to form large self-assembling bilayers, organized by electrostatic interactions, that cannot enter into the active site gorge of the enzyme (monomeric substrate depletion hypothesis); (b) at high concentration, the hydrolysis reaction products, P_1_ (choline) and/or P_2_ (benzoic acid), either do(es) not dissociate from the enzyme active center, causing product inhibition [36], or accumulate(s) inside the active site gorge where they/it may inhibit the entrance of new BzCh molecules (traffic jam hypothesis) or cause local pH decrease (benzoic acid).

Unlike the BChE-catalyzed hydrolysis of BzCh and related long-alkyl chain derivatives, that display damped [33,37] or stochastic [36] oscillations in the first minutes of steady-state hydrolysis in 0.1–0.2 M phosphate, pH 6.0 or 7.0, the AChE-catalyzed hydrolysis of BzCh was linear in 0.1 M phosphate buffer, pH 8.0. In the case of BChE, at low substrate concentration, oscillations at the beginning of the BChE-catalyzed hydrolysis of BzCh were interpreted in terms of slow equilibria between multiple molecular associates of BzCh molecules [37]. In the case of AChE, no oscillations were observed. Nevertheless, the first hypothesis (a) was checked by ^1^H-NMR. The ^1^H-NMR spectra of BzCh at different concentrations did not provide evidence either for dimer of BzCh molecules (π−cation interactions between choline and benzoic ring) or multimeric and micellar associates (Appendix A). The determination of the self-diffusion coefficients, by means of Fourier transform-pulsed gradients spin-echo (FT-PGSE) NMR, is known as a powerful tool for the characterization of supramolecular systems in solution. The self-diffusion coefficients (Ds) do not change with increasing BzCh concentration (Appendix A). Moreover, the results of tensiometry, spectrophotometry and DLS studies do not support the formation of such molecular associations. We see that BzCh does not decrease the surface activity on the air–water interfaces (Appendix A). The investigation of the concentration-dependent absorption spectra of the molecular state does not show any shift (Appendix A) or change in absorbance intensity with increasing of concentration (Appendix A). The DLS method revealed the formation of large structures with diameters of about 200 and 300 nm and a polydispersity index around 0.4 (Appendix A). Since spectrophotometry data about the solubilization of hydrophobic dye Sudan I did not reveal hydrophobic zones capable of solubilizing the dye, and data obtained from other methods denied the formation of self-assemblies, the DLS method cannot correctly reflect the formation of micelles. We may, therefore, refute the hypothesis that the formation of micelles impaired the penetration of single BzCh molecules into the active site gorge.

The second hypothesis (b) was checked by performing the steady-state hydrolysis of BzCh in the presence of choline and benzoic acid, the hydrolysis products P_1_ and P_2_, respectively, of BzCh hydrolysis. Moreover, ^1^H-NMR of BzCh solutions showed that a small fraction (2%) of BzCh was spontaneously hydrolyzed in choline and benzoic acid in highly concentrated solutions (Appendix A). The inhibitory action of choline on AChE was previously investigated by steady-state kinetic analysis and reported to be a weak reversible competitive inhibition (*K_i_* = 3.2 ± 0.4 mM) [33]. Moreover, there was no non-linear dependence of inhibition that could have suggested either allostery or partial inhibition. This linear and low inhibitory potency suggests that the presence of 2% choline in substrate solutions cannot significantly inhibit the enzyme.

Competing substrate kinetics of the AChE-catalyzed hydrolysis between BzCh and BzTC were performed in the presence of high concentrations (2; 3.2; 5 mM) of benzoic acid or choline and of both choline and benzoic acid. High concentrations of benzoic acid caused a slight decrease in pH buffer, e.g., pH = 7.6 in the presence of 5 mM benzoic acid. Such a pH decrease cannot explain the sudden drop of AChE activity observed beyond 600 μM BzCh (Figure 4A). Finally, the time-course hydrolysis of the AChE-catalyzed hydrolysis of BzTC, in the presence of high concentrations of choline, benzoic acid or both products (Figure 5), also showed that products do not impair the catalysis and cause only an increase in the time needed for the full completion of the substrate, acting like a reversible competitive inhibitor with an apparent overall *K_i_* = 0.5 ± 0.04 mM.

Such concentrations are much higher than the highest concentrations (<1 mM) of BzCh that we used in the steady-state kinetic experiments. However, if we state that the hydrolysis products of BzCh, namely choline (*P*_1_) and benzoic acid (*P*_2_), remain bound in the active site gorge where they accumulate, their local concentration rapidly increases in the 300 Å^3^ volume of the enzyme active site gorge.
(5)$$\left[P_{1}\right]=\left[P_{2}\right]=\left[H^{+}\right]=\left(\frac{k_{\mathrm{cat}}\left[E\right]\left[S\right]}{K_{m}+\left[S\right]}\right)\int_{0}^{t}dt$$

According to Equation (5), with [E] = 10^−8^ M and an apparent BzCh turnover of 72 min^−1^, at BzCh concentration 600 μM, much higher than Km, it would take about t = 800 min for the hydrolysis products to reach such a concentration if they accumulate in the active site gorge of AChE (volume = 300 Å^3^). Moreover, if benzoic acid (p*K*_a_ = 4.2) is dissociated at pH 8.0, and protons are also released with P_1_ and P_2_, then the local pH would drop below the enzyme p*K*_a_ if protons were not evacuated from the active site gorge; we observed only a modest pH decrease in spectrophotometer cuvettes at the highest BzCh concentrations. This situation is very different from the pH drop effect on enzyme velocity that was observed for the BChE-catalyzed hydrolysis of aspirin [6]. Thus, the time-dependent product accumulation hypothesis is not realistic.

Therefore, the unpredicted abnormal behavior of AChE at high concentrations of BzCh cannot be simply explained. A thorough investigation of the ChE-catalyzed hydrolysis of substrates at high concentrations is needed for all types of substrates to determine the catalytic mechanism over a large range of substrate concentrations. Regarding BzCh, the most likely explanation for the observed BzCh inhibition of AChE beyond 600 μM BzCh is that hydrolysis products, P_1_ or P_2_, remain bound to the CAS, leading to a catalytically unproductive ternary complex, SEP. Such an inhibitory phenomenon, due to the blockade of product dissociation from the CAS, was carefully investigated with AChE [38] and for haloalkane dehalogenase [39] with certain substrates. More detailed studies should be carried out on both ChEs with BzCh and also with the natural substrate acetylcholine. Under certain circumstances, i.e., the accumulation of acetylcholine or an exogenous substrate (poisonous or medicinal esters) in synaptic clefts and/or at neuromuscular junctions, the sudden inhibition of ChEs by a very large excess of substrate may have physio-pathological, toxicological or pharmacological significance and consequences.

### 2.3. Competing Substrate Kinetics of AChE

Competing substrate kinetics between low concentrations (0.25 mM) of BzTC as the reporter substrate, and increasing concentrations (from 0.5 to 1.2 mM) of BzCh as the blind substrate, was also performed in order to check the above-mentioned hypothesis. Kinetics were performed in 0.1 M phosphate buffer, pH 8.0 at 25 °C, according to [33]. The results showed that the time-course of competing progress curves are sigmoidal and that their plateau for maximum hydrolysis of the reporter substrate (BzTC) decreases with the concentration of competing substrate (BzCh) (Figure 6A). The sigmoidal shape indicates that the competing blind substrate is significantly hydrolyzed during the time course of the experiment [33,40]. The fact that *v_i_*/*v*_0_ curves vs. BzCh concentration does not reach 0 at high BzCh concentration (Figure 6B) also indicates that the inhibition is partial.

## 3. Discussion

The present results clearly show that the binding of a second positively charged substrate molecule on the PAS affects the acylation step when substrates are charged (thio)esters or an arylacylamide, e.g., acetyl-esters, benzoyl(thio)choline or arylacetyl-amides like ATMA. This confirms our hypothesis that *a* > *d* (cf. end of Section 4.3.1)

This effect can be the activation or inhibition of the acylation step. These mechanisms involve motions of both the Ω loop and the acyl-binding (ABP) loop. At high substrate concentration, the simulation of the *b* factor changes as a function of three variables *k*_2_/*k*_3_, *a* and *d* (cf Equation (7)), showing that, for the BChE catalysis of an ester like BTC (*b* = 3, cf Table 1) where *k_2_* and *k_3_* are partly rate-limiting, the contribution of the factor *a* is dominating (*a* = 4.5, *d* = 2.4). A similar conclusion can be drawn for AChE, where *b* < 1 with this substrate or ATC. However, in the case of an arylacylamide (*k*_2_ << *k*_3_), for both enzymes, the factor *a* is the sole contribution to *b*, i.e., *a* = *b*.

Several unanswered questions remain. In particular, it is not clear why both enzymes display opposite behavior with most positively charged esters but not with ATMA (Table 1). Site-directed mutagenesis studies on AChE showed, also, that mutations on F297 and around this position in the acyl-binding pocket (ABP) impact the value of the *b* factor in an opposite way [26,27,41,42]. In silico simulations, using molecular docking and QM/MM, should provide a definitive answer to this question.

We must point out that a thorough kinetic study showed that high concentrations of acetylcholine-, ATC- or a positively charged analog accelerate the decarbamylation of *Drosophila* AChE, carbamylated by a neutral carbamyl-ester (*k*_3_ >> *k*_2_). The acceleration of deacylation (decarbamylation) results from the binding of the second molecule (acetylcholine, ATC, substrate analog) at the rim of the active site gorge, i.e., near the PAS [43]. Our results, showing that *a* > *d* for carboxyl-esters (*k*_3_ ≈ *k*_2_) and arylacylamides (*k*_2_ << *k*_3_), do not follow this explanation. Indeed, in the case of the acceleration of the hydrolysis of a carbamyl-ester, *d.k*_3_ is increased by high concentrations of a positively charged ligand, because *k*_2_ >> *k*_3,_
*d*.*k*_3_ is always smaller than *a.k*_2_ (even if *a* is increased too).

This result also highlights that reversible and irreversible inhibitors of pharmacological and toxicological interest may interfere with the complex catalytic mechanisms of ChEs. Irreversible inhibitors like carbamates and organophosphates alkylate the CAS serine, then decrease the free active enzyme concentration; they may also bind reversibly to the PAS and modulate the reactivity of the CAS (cf. interaction of VX with the PAS of AChE [44] or acceleration of AChE inhibition by PAS ligands [45]) Ligands forming non-covalent complexes act as reversible competitive, non-competitive, mixed-type or uncompetitive inhibitors. Reversible inhibition is fast with most ligands (equilibrium is reached within microseconds). However, it can be slow, in particular with bulky ligands [46,47]. Exclusive binding on PAS makes ternary complexes (I_p_ES), determining uncompetitive inhibition. Reversible inhibition can be total (linear inhibition) or, less frequently, partial (hyperbolic inhibition) [48]. Inhibitors that bind on both CAS and PAS determine more complex reversible inhibition patterns. All reversible and irreversible inhibitors are either potent toxicants or important drugs used for the treatment of various diseases, in particular Alzheimer’s disease [49,50]. The 3D structures of numerous covalent conjugates and non-covalent complexes have been solved in the past 25 years. Although binding kinetics and inhibition mechanisms are still puzzling for certain of these molecules [46], adaptative conformational changes, upon the binding of various ligands [28], shed light on the complex interplay between the activity, binding and inhibition of ChEs. Thus, the effects of substrate/ligand binding to PAS on the acylation step of both ChEs has important functional implications with respect to the cholinergic mechanisms and pharmacological/toxicological actions of ChE ligands. Since the catalytic response of AChE and BChE to the binding of a second substrate molecule on the PAS are not systematically opposite, the response depends on the chemical structure of the substrate (Table 1) and its productive adjustment in the CAS (ES ⇄ ES^≠^), leading to enzyme acylation (EA). Thus, a simple kinetic analysis shows its limitation.

## 4. Materials and Methods

### 4.1. Chemicals

Acetylthiocholine iodide (ATC), butyrylthiocholine iodide (BTC), benzoylcholine chloride (BzCh) and dithio-bisnitrobenzoic acid (DTNB) were purchased from Sigma–Aldrich (Saint Louis, MO, USA) and benzoylthiocholine chloride (BzTC) from ICN (Tokyo, Japan). N3-(acetamido)-N,N,N-trimethylanilinium iodide (ATMA) was synthesized according to the method described by Johnson et al. [32]. Based on NMR spectra, purity of ATMA was >99%. 1H- and 13C-NMR spectral data of ATMA are in Appendix A. Stock solution of 0.1 M ATMA was in water. Stock solutions of ATC, BTC, BzCh and BzTC (0.1 M) were in water and stored at −20 °C. Echothiophate iodide was from Biobasal AG (Basel, Switzerland). Stock solution of 0.1 M echothiophate was in water and stored at −20 °C. Choline chloride was from Acros Organics (Geel, Belgium). Benzoic acid was from Vecton (Saint Petersburg, Russia). All other chemicals were of biochemical grade.

### 4.2. Enzymes

Human BChE tetrameric form (MW = 340 kDa), highly purified from human plasma Cohn fraction IV-4 [51], was a gift from Dr. O. Lockridge (UNMC, Omaha, NE, USA). The enzyme was diluted in 0.1 M sodium phosphate buffer, pH 7.0, to an activity of 45 units/mL with 1 mM BTC as the substrate at 25 °C (one unit corresponds to the number of micromoles of substrate hydrolyzed per minutes). The diluted enzyme was titrated according to Leuzinger [52], using echothiophate as the titrant. The active site concentration of this preparation was 1.9 × 10^−7^ M.

Highly purified (>95% pure) recombinant human AChE monomer (MW = 70 kDa) was in solution in 10 mM HEPES, pH 7.5, containing 10 mM NaCl [53]. AChE concentration was 14.7 mg/mL based on absorbance at 280 nm (for highly purified BChE, A280 = 1.7 corresponds to mass concentration of 1 mg/mL). The enzyme was stabilized by bovine serum albumin (1 mg/mL *w*/*v*). A second preparation, 53.4 mg/mL, was stabilized by a polysacharide, citric pectine at 1 mg/mL *w*/*v*. The active site titration of stabilized AChE preparations, using echothiophate as the titrating agent, provided active site concentrations of 1.1 × 10^−3^ M and 4 × 10^−2^ M, respectively.

During the titration processes, enzyme activity was checked using the method of Ellman [54] with 1 mM BTC in 0.1 M phosphate buffer pH 7.0 for BChE and 1 mM ATC in 0.1 M phosphate buffer pH 8.0 for AChE.

### 4.3. Steady-State Kinetic Analysis

#### 4.3.1. Rationale

For simulation of steady-state kinetics of substrate hydrolysis by AChE and BChE, substrates were chosen so that the acyl-intermediates were the same for each enzyme, i.e, same *k_3_* for each enzyme regardless of the substrate used. For hydrolysis of esters, both chemical steps, acylation and diacylation, are partly rate-limiting, i.e., *k*_2_ is of the same order of magnitude as *k_3_*. Then, Equation (2) can be re-written as:(6)$$k_{cat}=\frac{k_{2}}{1+\frac{k_{2}}{k_{3}}}$$

For the arylacetylamide substrate ATMA, acylation is the rate-limiting step, i.e., *k_2_* << *k*_3_ and thus, *k_cat_* = *k*_2_. This kind of substrate simplifies the analysis.

From Equation (4), it follows that at high substrate concentration (ester), *v = b*·*k_cat_*·[*E*]. The *b* factor may be regarded as a phenomenological composite variable resulting from the contribution of two components: “*a*” that alters the acylation rate (*a·k*_2_) and “*d*” that alters the deacylation rate (*d·k*_3_). Then, considering the catalytic constant at high substrate concentration as *b·k_cat_*, Equation (6) leads to the following expression for this catalytic rate constant for hydrolysis of a (thio)ester:(7)$$bk_{cat}=\frac{ak_{2}}{1+\frac{ak_{2}}{dk_{3}}}$$
and, for hydrolysis of an arylacylamide substrate like ATMA, *bk_cat_* = *ak_2_*.

Combining Equations (2) and (7) leads to:(8)$$b=\frac{a(1+k_{2}/k_{3})}{1+\frac{ak_{2}}{dk_{3}}}$$

When the mechanism obeys the Michaelis–Menten model, *b* = 1 and therefore, *a* = *d* = 1. On the other hand, when there is activation by excess substrate, *b* > 1, *a* > 1 and *d* > 1. When there is inhibition by excess substrate, *b* < 1, *a* < 1 and *d* < 1. Therefore, with partly rate-limiting ChE-catalyzed hydrolysis of charged substrates (*b* ≠ 1), the ratio *k_2_/k_3_* at high substrate concentration can be expressed by:(9)$$\frac{k_{2}}{k_{3}}=\frac{a-b}{a\left(\frac{b}{d}-1\right)}$$

However, because of the allosteric effects, caused by motion of both the Ω loop and the acyl loop upon binding of S_p_ (see Figure 1), it can be reasonably hypothesized that these effects are more pronounced on acylation than on deacylation. Thus, it is expected that *a* > *d*.

#### 4.3.2. Steady-State Kinetics of Substrate Hydrolysis

Steady-state of ATMA, BzCh and BzTC hydrolysis by AChE and BChE were performed at 25 °C, at optimum pH of enzymes, i.e., in 0.1 M sodium phosphate buffer, pH 8.0, for AChE and 0.1 M sodium phosphate buffer, pH 7.0, for BChE. The active site enzyme concentration in assays, [E]_0_, was 10^−8^–10^−9^ M for AChE and 5 × 10^−9^ M for BChE.

For ChE-catalyzed hydrolysis of ATMA, the concentration of ATMA ranged from 0.05 to 9 mM. Hydrolysis of ATMA was monitored by the absorbance change at 290 nm (ε_TMA_ = 1850 M^−1^ cm^−1^) [32]. For hydrolysis of BzCh, the BzCh concentration ranged from 1 to 800 μM. Hydrolysis kinetics of BzCh was monitored by recording the decrease in absorbance at 240 nm (the difference in the extinction coefficient between substrate and products, Δε, is 6700 M^−1^ cm^−1^ at 240 nm in phosphate buffer [37,55]). For the thioester BzTC, hydrolysis was followed according to the method of Ellman et al. [37,54] with 0.33 mM dithio-bis-nitro-benzoate (DTNB) as the chromogenic reagent, by recording the increase in absorbance at 412 nm of 5-thio-2-nitrobenzoate (ε = 13,300 M^−1^ cm^−1^), resulting from the reduction in DTNB by thiocholine, the substrate hydrolysis product P_1_. The concentration in BzTC ranged from 5 μM to 5 mM.

Kinetic runs were performed at least in triplicate and catalytic parameters were determined by weighted non-linear fitting of rate equations (Equations (1) and (4)), using Origin (Originlab Co., Northampton, MA, USA). Kinetic and binding parameters are provided with standard errors of the mean. Discrimination between steady-state kinetic models (Michaelis–Menten (Figure 1) vs. Webb model (Figure 2)) were made by using the statistical analysis of residuals proposed by Bartfai and Mannervik [56]. Principles of this procedure, thoroughly expanded by Cornish-Bowden [57], are given in Appendix A.

#### 4.3.3. Possible Inhibition of AChE-Catalyzed BzCh Hydrolysis by Reaction Products

These studies were only performed with AChE that displays a very low activity with BzCh as the substrate at pH 8.0 and 25 °C. At high concentrations of BzCh (above 500 μM), the possibility that AChE was inhibited by released hydrolysis products was considered. The reaction products P_1_ (choline) and/or P_2_ (benzoic acid) were added to the medium either for steady-state kinetic analysis or for time-course of competing substrates kinetics. At the same time, for kinetics in the presence of benzoic acid (p*K*_a_ = 4.2) possible pH decrease was controlled.

Steady-state kinetics

Steady-state kinetics of BzCh hydrolysis at 3 different concentrations, 100, 250 and 500 μM, was performed in the absence and presence of choline (product P_1_) at one concentration: 3.5 mM. Because the product P_2_ (benzoic acid) adsorption at 240 nm, study of the effect of P_2_ was performed by competing substrate kinetics (next section).

Time-course of competing substrate kinetics

Time-course of complete AChE-catalyzed hydrolysis of the reporter substrate (BzTC) at low concentration (0.25 mM, i.e., less than *K*_m_) was performed in the absence and presence of BzCh as the blind substrate at various concentrations (0.5, 0.8, 1, 1.2 mM). Certain kinetic runs were also performed in the presence of choline and/or benzoic acid, the hydrolysis products P_1_ and P_2_ of BzCh, at concentrations 2, 3.2, 5 mM. Analysis of progress curves was performed according to the method we previously developed [33,40].

### 4.4. ^1^H-NMR of BzCh Solutions

To check whether BzCh at high concentrations can form multiple associates from non-covalent dimers to different type of micelles, ^1^H-NMR spectra of BzCh chloride solutions from 0.25 to 50 mM in 0.1 M phosphate buffer, pH 8.0, were performed at 30 °C. ^1^H NMR spectra were recorded on a Bruker AvanceIII-500 spectrometer working at 500.1 MHz in ^1^H and 125.8 MHz in ^13^C experiments. Chemical shifts were reported in ppm relative to residual signals of protons of deuterated solvents. D_2_O-d6 were used as NMR solvents. (Appendix A).

### 4.5. Tensiometry of BzCh Solutions

Surface tension measurements of BzCh solutions were performed using the du Nouy ring detachment method (Kruss K6 Tensiometer, Hamburg, Germany). Briefly, the spherical ring was placed parallel to the air/solvent interface. Between two surface tension analyses, the ring was cleaned with ultra-purified water, followed by soaking in ethanol and drying. Temperature was maintained at a constant at 25 °C during all measurements. (Appendix A).

### 4.6. Dynamic Light Scattering

Size and polydispersity index of BzCh solutions were determined by dynamic light scattering (DLS) measurements, using the Malvern Instrument Zetasizer Nano (Worcestershire, UK). Measured autocorrelation functions were analyzed by Malvern M1 DTS software v.7.13, applying the second-order cumulant expansion methods. The effective hydrodynamic radius (*R_H_*) was calculated according to the Einstein–Stokes equation *D = k_B_T/6πηR_H_*, where *D* is the diffusion coefficient, *k_B_* is the Boltzmann constant, *T* is the absolute temperature and *η* is the viscosity. The diffusion coefficient was measured at least in triplicate for each sample. The average error of measurements was ±4%. (Appendix A).

### 4.7. UV Spectrophotometry and Dye Solubilization

The concentration-dependent absorption spectra of BzCh solutions were measured using PerkinElmer λ35 (PerkinElmer Instruments, Waltham, MA, USA). Solubilization of the dye (Sudan I) was performed by adding an excess of crystalline Sudan I to BzCh solutions. These solutions were allowed to equilibrate for about 48 h at constant temperature (25 °C), followed by filtration. UV absorbance was measured at 485 nm (for Sudan I). Quartz cuvettes (1 cm-path) containing sample were used. (Appendix A).

## 5. Conclusions

The purpose of this work was to interpret the phenomenological *b* factor in terms of its acylation vs. deacylation contributions to the catalytic constant of ChEs at high concentrations of positively charged substrates. The magnitude and opposite values of *b* between AChE and BChE indicated that the productive adjustment of substrates in the active center depends on the motions of both the Ω and the acyl-binding loops, resulting from the occupancy of the PAS by a second substrate molecule. Remembering that the active site gorge of AChE is 300 Å^3^ against 500 Å^3^ for BChE, the poor catalytic hydrolysis efficiency of AChE against the bulky ester benzoylcholine illustrates the importance of the fine adjustment of the substrate acyl moiety in the acyl-binding pocket. Bulkier esters are not hydrolyzed by AChE while they are substrates of BChE. This property of BChE, capable of accommodating large molecules in its CAS, has important toxicological and pharmacological implications for the metabolism of ester-containing drugs.

Now, to understand the intimate mechanism of the activation versus inhibition of ChEs at high substrate (or ligand) concentrations, a thorough analysis of the catalytic pathway, including the cross-talk between PAS, CAS and ABP, is needed. For this purpose, QM/MM simulations of substrate hydrolysis should confirm that *b* depends on the effect of PAS occupancy on the dissociation of the bound substrate tetrahedral intermediate into the acetylated enzyme and alcohol/phenol product P_1_. Moreover, in silico studies should shed light on the effect of the size of the acyl moiety of the substrates on the stabilization of ES^≠^. Finally, these works are expected to support the recent findings of Radic’s group on X-ray and neutron diffraction/scattering and on MD simulations of AChE conjugates [28,29,58,59].

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
