# Peer review of "Activation/Inhibition of Cholinesterases by Excess Substrate: Interpretation of the Phenomenological b Factor in Steady-State Rate Equation"

_ijms, 2023, doi:10.3390/ijms241310472_

Round 1

Reviewer 1 Report

We consider the manuscript very pertinent to the readers of this Journal's “Biochemistry” Section.

The study is the continuation of the previous work of P. Masson´s team, aiming for clarification of cholinesterase´s (AChE and BChE) mechanisms.

The objectives of this work are clearly stated and comprehensively justified.

The introduction briefly covers old and new references and perfectly integrates the theme's main aspects.

The core experimental design of the chemistry part of the manuscript seems carefully elaborated and meticulously presented.

This article is well written, with a good organization of the contents, only minor comments will be made concerning the estimation of kinetic parameters.

Regarding the discussion of the results, we found it nicely done. A very nice graphic pic of BChE’s active site (Figure 1_L100-101) was added to the Introduction to help the understanding of the conformational binding changes, but no authorship was added to that picture.

 Dear authors, please include (Comment #1) the authorship of the picture site (Figure 1_L100-101).

Still, about the discussion, various graphs were added to increase the understanding of the discussed theme and clarify the reasoning, but we didn´t find any results from the estimation of parameters to give the proper weight of a statistical tool to support the conclusions. The material and methods section referred to a kinetic data fitting using Non-Linear Regression (L468-470).  Dear authors, please include (Comment #2) the estimates for eq 1 and 4, their SE, and the residual analysis plot.  If the quality of estimates is poor, the conclusions lack statistical support.  

The preferred expression for an estimate and its precision is the mean and the 95% confidence interval (the range of values about 2 SEMs above and below the mean) [“Standard deviations and standard errors” _ DOI: 10.1136/bmj.331.7521.903]. It be noticed that “mean±SD” and “mean±SEM” are not the same. Please provide the precision of estimates and the residual graph (residuals vs predicted values) for the predictive models.

The description of the enzyme kinetic studies was sufficient to guarantee reproducibility.

Specific Comment:

#__ “Conclusions” _ The conclusion item is not mandatory for this journal, however, a short concluding text is especially important when the discussion is complex, as seems to be the case. Dear authors, please provide, if possible a concise text of the main achievements in this study. 

Author Response

Answer to reviewer 1

We thank Reviewer 1 for his comments and recommendations.

  • About Fig 1, there is no authorship. This figure was made by using the PDB coordinates of human BChE x-ray structure (PDB: 1PO1) and using PyMOL for visualization of the molecular structure. We added this mention this in the Figure 1 legend of the revised manuscript. Note that P. Masson was co-author of the resolution of the 3D structure of the enzyme (ref 18: Nicolet et al., JBC, 2003).

  • About statistical analysis of data, parameters were calculated using Origin. Experiments were performed at least in triplicates, regression analysis was weighted and standard errors are SEM. We added this in lines 487-489.

No, residual analysis plots were added because data were very accurate. Visual inspection of typical plots, rate vs substrate concentration (cf Figs 3 and 4), are clear on that point.

  • About the conclusion. We dissociated the chapter “discussion and conclusion” (item 3.) and wrote a separate conclusion (item 4.). For this purpose, the conclusion contains two paragraphs. The first one (line 384-395) is completely new, it summarizes the main achievement of this study and emphasizes the differences between AChE and BChE. The second paragraph was moved from the end of discussion and modified according to your recommendations. It gives clues about what will be done in the next step of this work.

All changes are marked in red in the revised manuscript.

Patrick Masson

Reviewer 2 Report

The manuscript by Mukhametgalieva et al. entitled “Activation/Inhibition of cholinesterases by excess substrate: interpretation of the phenomenological «b »factor in steady-state rate equation”, reported an accurate in vitro and in silico analysis regarding the parameters that influence the overall rate of the reaction, i.e. the rate limiting step. Since the study concerns cholinesterases, enzymes fundamental in the transmission of nerve signals, it may be relevant in elucidating particular aspects of these enzymes. In other words, the proposed topic is of interest. Moreover, the manuscript is well written, and the experiments are well conducted. However, my opinion is to consider this manuscript for publication after minor revisions.

 I suggest the authors to revise the manuscript by following the points reported below:

-       Page1, line 20; “b” is in Italic.

-       In the text the references number, figures and scheme are highlighted, please correct them.

-       Page 4, line 164; “b” is in Italic.

-       Page 8, line 289; do the authors mean [E]=10-8 or [E]=10-8?

-       Page 10, line 335; it is reported “Table I”, do the authors mean “Table 1”?

-       Page 10, lines 338-339; rewrite the phrase removing the referment to the parenthesis “(S. Luchkchekina et al., …). Do not report unpublished results.

-       Page 11, line 377; it is reported “Table I”, do the authors mean “Table 1”?

-       Page 11, lines 384-385; do not report results or conclusions from unpublished data.

Best regards

Author Response

Answer to Reviewer 2

We thank Reviewer 2 for his comments and recommendations. We corrected typos and rewrote pages 10 and 11 sentences in compliance with his remarks.

All changes are marked in red in the revised manuscript.

Patrick Masson

Round 2

Reviewer 1 Report

Dear authors, we thank you for your effort in responding to comments. However, some statistical concerns remain regarding the statistical assumption that the rates have a uniform coefficient of variation (non-linear regression assumptions ought to be assured!)

Comment #1_when the authors were asked:

"Dear authors, please include (Comment #2) the estimates for eq 1 and 4, their SE, and the residual analysis plot. If the quality of estimates is poor, the conclusions lack statistical support."

The author's response was:

"No, residual analysis plots were added because data were very accurate. Visual inspection of typical plots, rate vs substrate concentration (cf Figs 3 and 4), are clear on that point."

There seems to be some confusion of concepts here, which we will try to reason out.

The visual inspection (as stated by the authors) of the fitted data only reflects the adjustment of the data to the considered model. Residual graphs allow us to obtain information beyond the blind adjustment and have been vastly recommended by modeling enzymologists such as Athel Cornish Bowden (well-known author and one of the most experienced researchers on the topic). The vital importance of graphical analysis of residuals is discussed at length in the Book "Fundamentals of Enzyme Kinetics" (see chapter 15 4th Edition) [ISBN: 978-3-527-33074-4] by Athel Cornish-Bowden.

Residual plots are used to show evidence of possible systematic errors. It is noted that without the residual plot, the wrong model fit appears to be excellent, and the systematic nature of the errors becomes obvious with just the residual plot, as is shown didactically in Figure 2.8; pag22 of the book "Practical Enzymology" by Hans Bisswanger [2nd Edition ISBN: 978-3-527-65924-1].[vide pdf attached]

General programs like Origin, with simple enzymatic modeling algorithms, are very practical tools for non-enzymologists, but they are not sensitive enough to detect these particular points in the estimation of enzymatic biological mechanisms.

The fact that the data have a good fit does not imply the absence of errors that could make it unfeasible to accept the parameters as estimates.

Upon further analysis of Figure 3 [L200-201], this issue becomes even more demanding, since there appears to be a discontinuity at the experimental points between the 1000 and 3000 uM [BzTC] which corresponds to half the maximum rates region and therefore the estimation region of Km.

#1__So, dear authors, please include the graphical analysis of residuals for estimates under discussion and explain why there is a lack of experimental data around the Km. To be noticed that Figure 4B there is unbalanced data too above and below Km.

Question #2

Figure 4 it´s supposed to represent the fitted Michaelis Menten equation 1.  The hyperbola is expected to be defined by YY axis representing initial velocity data (μMmin-1), as a function of substrate concentration (μM) [https://www.beilstein-institut.de/en/projects/strenda/guidelines/]

#2__Since the optical density variation (ΔA) cannot be defined directly as the concentration of the compound under study, how was the initial velocity defined, and how were defined the weights used in the regression? 

Author Response

Answer to Reviewer 1

#1: We thank Reviewer 1 who pointed out the importance of residuals analysis for discrimination between catalytic models. We were aware about this statistic analysis method from Cornish-Bowden classic textbook but we never used it. Thus, we performed the analysis according to the method developed by the discoverers (Bartfai and Mannervik, new ref 58). Details and figures are in Supplementary file, and conclusions are in the manuscript.

#2: initial velocity (vi) is d(P)/dt at t=0. In steady-state kinetic run, rate is linear for more than 5 min, and in general we record kinetics for 3 min. However, initial velocity is important in analysis of complete time-course of kinetics (integration of rate equations). Accurate determination of vi was needed for building plots in Figs 5B and 6B. Theoretical and experimental details are reported in ref 39 and 40.

Weights used in regression are reciprocal of variance of recorded rates. They were automatically incorporated in Origin.

All new changes in manuscript and supplementary file are written in green.

Sincerely yours,

Patrick Masson

Round 3

Reviewer 1 Report

We appreciate the efforts of the authors to clarify the requested points.

We found that the residual plots show systematic errors in the estimates made in the regression. This implies that the discussion and conclusions about the kinetic parameters and mechanism of the enzyme ought to be done with great care. Therefore, dear authors, please call attention to readers that more detailed studies should be done on the enzyme mechanism.

In addition, the Michaelis Menten graphs must be presented with the initial velocities in units of [P] micromolar(or M) per unit of time (min or s). Therefore, the variation in optical density must be ensured by the dilution factor and the extinction coefficient of the product.

Author Response

Response to Reviewer 1

All changes in text, figures and figure legend (Fig 4) are written in blue.

  • The residuals analysis shows no errors up to 600 microM BzCh. The distribution of residuals is horizontal in a narrow envelope. Thus plots of residuals show ideally scattered points.

Otherwise, analysis of data up to 800 microM BzCh generates weird plots of residuals, indicating lack of fit, i.e. both catalytic models, Michaelis-Menten and Webb-Radic are wrong in this substrate concentration range. In the results, below Fig. 4, we point out this problem. We also emphasize this critical issue at the end of section 2.1 of results.  An unexpected phenomenon occurs between 600 and 800 microM BzCh. Understanding this kinetic anomaly is one of our priorities, and of course, more detailed studies will be undertaken

  • Graphs of Figs 4 were presented using micromoles of substrate hydrolyzed per min i.e, micromoles of products P released per min rather than absorbance change a 240 nm per min.

Patrick Masson